# Exploring Physical Activity in Children and Adolescents with Disabilities: A Bibliometric Review of Current Status, Guidelines, Perceived Barriers, and Facilitators and Future Directions

**DOI:** 10.3390/healthcare12090934

**Published:** 2024-05-01

**Authors:** Ye Ma, Mengjiao Liu, Yuwei Liu, Dongwei Liu, Meijin Hou

**Affiliations:** 1Research Academy of Grand Health, Faculty of Sports Sciences, Ningbo University, Ningbo 315211, China; maye@nbu.edu.cn (Y.M.); 2211040074@nbu.edu.cn (M.L.); 2Auckland Bioengineer Institute, University of Auckland, Auckland 1010, New Zealand; yilu378@aucklanduni.ac.nz; 3School of Information Technology and Artificial Intelligence, Zhejiang University of Finance and Economics, Hangzhou 310018, China; 4National Joint Engineering Research Centre of Rehabilitation Medicine Technology, Fujian University of Traditional Chinese Medicine, Fuzhou 350122, China; 5Key Laboratory of Orthopaedics and Traumatology of Traditional Chinese Medicine and Rehabilitation, Fujian University of Traditional Chinese Medicine, Fuzhou 350122, China

**Keywords:** physical activity, disability, children, adolescents, bibliometric analysis

## Abstract

Background: Physical activity contributes to both physiological and psychosocial benefits for children and adolescents with disabilities. However, the prevalence of physical inactivity is notably higher among disabled young people compared to their healthy peers. Despite this, there is a lack of constructed knowledge structure, evolutionary path, research hotspots, and frontiers in studies related to physical activity in young people with disabilities.Methods: The literature related to the research of physical activity in children and adolescents with disabilities was retrieved from the core collection of the Web of Science. The annual publication numbers and the timing, frequency, and centrality of the co-occurrence network with respect to journals, countries, institutions, authors, references, and keywords were analyzed. Additionally, clustering analysis and burst analysis were performed on the references and keywords. All analyses were conducted using CiteSpace. Results: A total of 1308 related articles were included. The knowledge structure of research on the physical activity of disabled children and adolescents, including annual publication numbers, influencing journals, countries, institutions, authors, references, and keywords along with their respective collaborative networks, has been constructed. Furthermore, the research foundation, current hot topics, and research frontiers have been identified by analyzing references and keywords. Conclusions: Current research hotspots include interventions, therapies, and programs aimed at enhancing specific skills, as well as addressing the satisfaction of competence to improve motivation and the effectiveness of physical activity. There is also a focus on the development of scales for quantitative studies. Future directions may be toward personalized interventions or programs to enhance physical activity levels among youth with disabilities.

## 1. Introduction

Children and adolescents living with disabilities form a highly varied group, encompassing a range of life experiences. They exist in communities globally and may develop a disability either at birth or during their childhood or adolescence [1]. UNICEF reported a global count of 240 million children affected by disabilities [2]. Ensuring regular physical activity is crucial for the well-being of disabled children and adolescents, positively impacting their physical and psychological health, and therefore promoting their quality of life [3,4,5].

Physical activity contributes to improved physical fitness, such as cardiorespiratory fitness, cardiometabolic health (including blood pressure, dyslipidemia, glucose, insulin resistance, etc.), muscle strength and endurance, bone health, and flexibility [3,4]. Moreover, participation in physical activity also enhances an individual’s motor skills and coordination, resulting in better physical functioning and independence [6,7]. Heightened engagement in physical activity and enhanced fitness levels offer additional benefits for youth with disabilities. These benefits lower the risk of chronic diseases in adulthood, such as diabetes and heart disease. Furthermore, they contribute to the mitigation or elimination of secondary conditions like obesity, weakness, fatigue, and mobility issues [5]. Therefore, increased physical activity reduces the dependency on personal assistance for performing the activities of daily living.

Physical activity also significantly enhances the psychological health of disabled youth, promoting positive mental health, reducing anxiety and depression symptoms, and boosting self-esteem among youth with disabilities. In sports or recreational activities, they experience achievement, build resilience, and develop coping mechanisms. Additionally, physical activity fosters social interaction, friendships, and a sense of belonging, acting as a catalyst for social integration. Inclusive programs contribute to the development of social skills, empathy, and understanding, breaking down stereotypes and fostering a culture of acceptance [3,4,5].

Despite the physiological and psychosocial health benefits, the rate of physical inactivity among youth with disabilities is much higher than in youth without disabilities [5,8,9]. The youth with disabilities spend significantly less time in “habitual physical activities”, which includes occupational activities, leisure activities, organized and spontaneous sport games [10,11,12], and considerably more time on sedentary times [5,9,10,12,13].

Bibliometric analysis is a popular and rigorous method for exploring and analyzing large volumes of scientific data via mathematical and statistical methodologies [14]. Scholars are able to understand the characteristics and structures of a field’s development and uncover emerging trends and hotspots in article and journal performance; collaboration patterns between countries, institutions, or authors; reference and keyword clustering and burst analysis; and so forth [14]. Therefore, bibliometric studies provide researchers with a comprehensive overview, aiding in identifying current research gaps, deriving new research ideas, and positioning contributions to the field [14,15].

Researchers have investigated the association between physical activity and sleep [16], the impact of physical activity on the health of old adults [17], the influence of physical activity and a healthy diet on physical health, mental health, and well-being [18], the physical activity-related research of autism [19], the relationship between physical activity interventions and cognition in children and adolescents [20], and physical activity studies after the COVID-19 pandemic [21] using bibliometric analysis. With these analyses, researchers can construct a general research landscape, gain valuable insights into those areas, and monitor research development and the patterns of publication.

Since 1995, there have been numerous studies focusing on the physical activities of children and adolescents with disabilities. Those studies cover broad aspects, including physical activity guidelines [3,4,11], the current status of physical activity participation [5,10,12,22,23,24], perceived barriers and facilitators as well as the factors associated with physical activities [25,26,27,28,29,30], physical activity promotion [25,26,27,28,29,30] and so forth in disabled youth. However, there is no bibliometric study on this topic, which comprehensively constructs the knowledge structure, evolutionary path, research hotspots, and frontiers in physical activity research in disabled youth.

Therefore, the aim of this study is to conduct a comprehensive bibliometric review of the publications on physical activity in children and adolescents with disabilities from 1995 to 2023, providing a clear view of the current research structure, identifying the foundation research, highlighting the current research hotspots and exploring the developmental trends.

## 2. Materials and Methods

### 2.1. Data Acquisition

The Web of Science (WoS) was chosen as the search engine due to its internationally recognized database, known for maintaining the highest quality standards and being widely accepted and utilized for scientific publication analysis [31,32]. The WOS core collection citation database offers a unique feature of citation counts, enabling the quantification of the relative importance of articles from a large pool through objective measures [33]. The WOS has been extensively utilized for bibliometric analysis [17,18,19,20,21,33,34]. Furthermore, the bibliometric tool employed in this study is CiteSpace, originally designed for the WoS database.

The data of this study were collected from the WoS core collection in January 2024. The data search strategy was as follows: Topic = (“physical activity” AND (“disabled” AND (“children” OR “adolescents” OR “youth”))). As the earliest full-text-available literature specifically focusing on studies of physical activity among disabled youth was published in 1995, the retrieval period time was set from January 1995 to December 2023.

The literature screening process and inclusion criteria are demonstrated in Figure 1. The initial search was conducted in the WoS database, the world’s largest academic journal database, using the aforementioned search term, and 3404 publications were identified. An initial screening was performed using the WoS filter to make sure that the selected publications are research articles or review articles, written using the English language, indexed by the core collection of the WoS, and published in the period of January 1995 to December 2023. After the initial screening, 2938 publications were identified. Then, a manual screening was conducted by two independent individuals by checking the title, abstract, and main content to make sure that the records fitted the inclusion criteria and also worked on the physical activities of children or adolescents with disabilities. In total, 1308 publications were included for the bibliometric analysis, including 1033 research articles and 172 review papers.

### 2.2. Data Analysis

For the bibliometric analysis, the 1308 retrieved articles were first exported as plain text files, including both publication records and citations. Subsequently, those files were converted to an executable format using the “Data/Import/Export” function of CiteSpace (V6.2.R6, 64 bits).

CiteSpace is a multivariate, time-sharing, Java-based software used for analyzing a visualized citation network, which is widely used and accepted in bibliometric studies. CiteSpace was developed by Professor Chaomei Chen from the College of Computing and Informatics at Drexel University [35].

We identified annual publications and the timing, frequency, and centrality of the co-occurrence network of journals, countries, institutions, authors, references, and keywords. The optimal parameters for the bibliometric analysis were selected through iterative experiments with various parameter settings, evaluating the resulting network structure (such as modularity, silhouette values, and the number of clusters). This process aimed to achieve the most informative and visually clear representation of the research landscape [36]. The parameters used in CiteSpace were: time slice (1995–2023), the number of years per slice (4), term source (all selected), node type (selected one at a time), selection criteria (first 50 objects), g-index (K = 10), pruning (None) and visualization (cluster view-static).

The generated knowledge maps employ node-and-link diagrams to depict networks. Nodes represent authors, countries, institutions, references, and keywords, respectively. Within the co-occurrence plot, node size reflects keyword frequency, and colors denote distinct years. Links indicate cooperative, co-occurrence, or co-citation relationships, with larger nodes denoting a higher frequency. Line thickness represents co-occurrence intensity. The purple circle signifies centrality (>0.1), identifying pivotal nodes within a domain [37].

In addition, clustering analysis and burst analysis were conducted on references and keywords. The most important papers were identified in terms of the cited frequency, the strength of citation burst, and network centrality. These papers were identified as the foundation papers of the studies on physical activity in disabled youth. The co-citation clustering using the log-likelihood ratio strategies, with the selection type of “title word” and “keyword” for reference and keywords, respectively, was carried out. The clustering performance is measured by the modularity value (Q-value) and the weighted mean silhouette value (S-value), with the Q-value > 0.3 and the S-value > 0.5 indicating reasonable clustering [36]. Specifically, the modularity value measures the strength of the division of a network into clusters, with a high modularity value indicating that the network is well-divided into distinct, densely connected clusters with sparse connections between them [34,36,37,38]; the mean silhouette value is a measure of how well each element (i.e., reference and keywords) fits into the cluster it has been assigned to [36,37,38].

## 3. Results

### 3.1. Analysis of Annual Publication

A total of 1308 records were included in this study. The number of publications per year is presented in Figure 2. The graph illustrates a substantial increase in the number of publications over the past two decades, with some noticeable fluctuations. The number of publications can be fitted using a power function $y=0.0007716\times \left(x^{3.689}\right)$, in which y represents the number of publications and *x* represents the years since the first related research was published in 1995. The fitted curve shows excellent performance with an R^2^ of 0.9755 and a root-mean-squared error of 8.958. It is evident that since the research on physical activity in children or adolescents with disabilities was published in 1995, there have been only a few publications in the first decade, with less than ten publications per year.

Since the year 2007, there has been credible growing interest and research on the physical activity of disabled children and adolescents. There are mainly three sub-periods in the increase in booming research. The first one is from 2007 to 2011, the annual publication increased from 10 to around 30. The following is from 2012 to 2019, the annual publication is increased from 30 to 91. Since 2020, there have been more than 100 related research published per year after the COVID-19 pandemic. In 2023, the number of related articles published reached 210, showing the remarkable growing interest and research conducted on physical activity enhancement for disabled children and adolescents.

According to reports from the World Health Organization, the disability prevalence rose from approximately 10% in the 1970s to around 15% in 2011 and 16% in 2023 [39,40]. The increasing global prevalence of disability may be due to population aging, the widespread occurrence of chronic diseases, and advancements in disability measurement [39,40].

### 3.2. Analysis of Journals

Table 1 displays the top ten most productive journals with the highest volume of research on the physical activities of disabled children and adolescents. The leading journal is *Adapted Physical Activity Quarterly*, contributing 97 published papers out of a total of 370 publication sources. Categorized in rehabilitation and sports science in the *Science Citation Index Expanded* (SCIE), its primary focus is to provide the latest studies on physical activity for disabled individuals. Following closely are the second and third most productive journals, *Disability and Health Journal* and *Disability and Rehabilitation*. Both are multidisciplinary international journals advancing knowledge in disability, with a specific aim to promote health and rehabilitation. These two journals are both indexed by the SCIE and the *Social Science Citation Index* (SSCI).

The top co-occurrent journals related to the topic of physical activities for disabled children and adolescents (see Table 2) differ from the list of top productive journals (see Table 1). The most frequent co-occurring journal is *Research in Developmental Disabilities*, cited 625 times. This journal predominantly publishes empirical studies associated with developmental disabilities. Notably, the centrality of four cited journals, including *Disability and Rehabilitation*, *Developmental Medicine and Child Neurology*, *Adapted Physical Activity Quarterly*, and *Pediatrics*, has reached a centrality threshold of 0.10. This indicates that these four journals play a pivotal role in the area of the physical activity studies of disabled children and adolescents. The registered regions of all journals in Table 1 and Table 2 are from the United States and England, except for the *International Journal of Environmental Research and Public Health*.

The retrieved journals’ Web of Science Categories span both natural and social sciences, as depicted in Figure 3. The leading category is “Rehabilitation” with 566 records. Following closely are “Pediatrics”, “Public Environmental Occupational Health”, and “Sports Sciences”, each boasting more than 200 records.

### 3.3. Analysis of Countries

Table 3 and Figure 4 present the top 10 countries and the collaboration network in the research on physical activity for disabled children and adolescents, respectively. Figure 4 illustrates 74 nodes and 364 links, highlighting the countries’ contributions and cooperation in this field. The United States leads in all countries with a co-occurrence frequency of 495 times, followed by Canada, Australia, England, and Mainland China with 217, 170, 82, and 71 times, respectively.

In the top ten countries, the centralities of the United States, England, the People’s Republic of China, Australia, and Ireland exceed 0.10. The United States and England hold the highest centralities at 0.26 and 0.22, respectively. Despite a smaller number of frequencies, Belgium, Saudi Arabia, South Africa, and Romania also exhibit centralities exceeding 0.10, indicating a notable influence.

### 3.4. Analysis of Institutions

Table 4 outlines the top ten institutions in terms of the co-occurrence frequency. Along with that, the centrality and the year of first publication are also demonstrated. Figure 5 illustrates the collaboration map of institutions using co-occurrence analysis from CiteSpace.

It is noteworthy that the leading institutions are predominantly universities, with the exception of the *Utrecht University Medical Center*. The *University of Toronto* holds the highest number of frequencies among the 159 contributing institutions in the field of physical activity among young people with disabilities. Their first related paper was published in 2007, and the centrality is 0.12, indicating a dominant position in this field. Following closely are *Old Dominion University* and the *University of Queensland*, with the frequencies of 35 and 33 times, and centralities of 0.15 and 0.10, respectively, signifying their leading roles in research on physical activity among disabled young people.

The *Utrecht University Medical Center* and *Utrecht University*, both ranking fifth with 32 times of co-occurrence, exhibit low centrality (0.01), indicating a tendency to conduct independent research. Interestingly, the University of Sydney and the University System of Ohio, though not ranking in the top ten institutions based on co-occurrence frequency, have high centralities (0.11 and 0.10). This suggests their inclination to collaborate with other institutions and act as central hubs in their research networks.

### 3.5. Analysis of Authors

The top five productive authors in the field of physical activity for disabled children and adolescents are Justin A. Haegele (30 publications), Nora Shields (23 publications), Cindy H. P. Sit (20 publications), Kelly Arbour-nicitopoulos (18 publications), and James H. Rimmer (17 publications).

An analysis of the collaborative network of authors based on 1308 papers revealed 171 nodes and 252 links, indicating that these articles were written by 171 authors. The collaborative network of authors and cited authors is illustrated in Figure 6 and Table 5, and it differs from the top five productive list. In terms of cited frequency, the top author is Justin A Haegele (29), followed by Nora Shields (18), and Kelly P. Arbour-nicitopoulos (17). Regarding centrality, Kelly P. Arbour-nicitopoulos has the highest centrality of 0.19, followed by Kwok Ng (0.17), and Amy E. Latimer-cheung (0.17).

Collaborative groups with interconnected relationships exist among these authors, along with individual authors scattered throughout the network. The figure showcases prominent collaborative networks with five or more publications, representing influential research groups in the field. Additionally, the graph reflects social relationships between the authors, with many authors working in relatively stable partner groups, often comprising two or more core authors per group.

Kelly Arbour-nicitopoulos [41,42] with the fourth-largest number of publications and the highest centrality, emerges as the leading researcher in the field of physical activity for disabled children and adolescents. Kelly collaborates closely with authors such as Amy E. Latimer-cheung, Amy C. Mcpherson, Patricia Tucker, Kathleen A Martin, Salome Aubert, and Kwok Ng. Their first research, published in 2015 [43], qualitatively investigated the parental psychosocial determinants supporting sport participation for adolescents with mobility impairment. The focus of Kelly and colleagues lies primarily on sports, particularly inclusive out-of-school time physical programs, aiming for the physical skill development, social skills, and cognition of children or adolescents with physical or sensory disabilities [41,42,44].

### 3.6. Analysis of Reference and Burst Reference

Co-citation, cluster, and burst analysis were conducted on the references. The co-citation reference network and clustering diagram are demonstrated in Figure 7 and Figure 8. The reference bursts are displayed in Figure 9. The fifteen most important papers accounting for co-cited frequency, burst strength, and co-citation network centrality were identified (see Table 6).

The co-citation network comprises 252 nodes and 840 links. The reference with the highest centrality is “Differences in physical activity among youth with and without intellectual disability”, published by Einarsson et al. [10] in the journal *Medicine and Science in Sports and Exercise* in 2015.

The performance of reference co-citation clustering was evaluated using Q-value and S-value. As indicated in Figure 7 (bottom), the modularity Q is 0.7341 (>0.5), and the weighted mean silhouette S is 0.92 (>0.5). These values indicate that the clustering is reasonable, and the homogeneity of the clusters is acceptable. The top three clusters identified are intellectual disability, cerebral palsy, and physical activity.

### 3.7. Analysis of Keywords

The keyword network map (see Figure 10) and the cluster diagram (see Figure 11) reveal the research hotspots in the area of physical activities for disabled children and adolescents. The map comprises 269 nodes and 1604 links, with the identification of the top 50 most frequently cited keywords. In terms of the centrality and co-occurring frequency, the top four keywords are “cerebral palsy”, “physical activity”, “young children” and “adolescents”.

The top seven clusters are demonstrated in Figure 11, each containing more than twenty keywords. These clusters include “health promotion” (cluster 0), “obesity” (cluster 1), “mental health” (cluster 2), “leisure activity” (cluster 3), “motor skill” (cluster 4), “cerebral palsy” (cluster 5), and “quality of life” (cluster 6).

The keyword citation burst analysis, indicating rapid changes in keywords over time, highlights frontier research topics and key areas (see Figure 12). The top five keywords with the highest strength burst are “recreation” (8.75), “mental retardation” (8.02), “gross motor function” (7.55), “energy expenditure” (6.84), and “secondary conditions” (6.54). The top five keywords with the longest duration of burst are “body composition” (1995~2010), “recreation” (2007~2018), “mental retardation” (2003~2014), “energy expenditure” (1999~2010), and “walking” (2003~2014). Notably, keywords with the strongest citation burst in the past three years include “skills”, “competence”, and “scale”.

## 4. Discussion

The research presents the first bibliometric review of the studies related to physical activity in children and adolescents with disabilities. Through a systematic screening of relevant publications in the core collection of the Web of Science database from 1995 and 2023, a total of 1308 papers were identified for a comprehensive bibliometric review. The knowledge structure was developed, including annual publications, productive and influential journals, leading countries, and institutions in the research of physical activity in disabled youth. We also identified research foundations by analyzing reference co-citations and references with the strongest citation bursts. The current research hotspots and future research frontiers were clarified through keyword co-occurrence analysis, clustering, and burst analysis.

Since 1995, publications on the studies of the physical activities of disabled children and adolescents show a substantial increase during the past 29 years. We found that the related studies identified in the Web of Science database amount to only an average of 45 papers per year. The publication trend can fit into a power function. Since 2007, there has been a growing interest, with three sub-periods of increased research: 2007–2011 (10 to around 30 publications annually), 2012–2019 (30 to 91 publications annually), and post-2020 (over 100 publications per year, reaching 210 in 2023), reflecting a remarkable surge in research on enhancing physical activity for disabled children and adolescents. With the rapid advancement of technology, new technologies and theories emerge extensively and exponentially. However, in comparison, research on physical activities for disabled children and adolescents is severely lacking.

### 4.1. Knowledge Structure in Terms of Journals, Countries, and Institutions

#### 4.1.1. Knowledge Structure in Terms of Journals

The studies of physical activity in disabled youth are highly interdisciplinary, which encompass more than ten areas of both natural and social sciences. Researchers should seek interdisciplinary collaborations to bring diverse expertise and perspectives to this topic. The top three productive journals are *Adapted Physical Activity Quarterly*, *Disability and Health Journal*, and *Disability and Rehabilitation* (see Table 1), which are highly specialized. Researchers could target these specialized journals to ensure their work reaches the most relevant audience and has maximum impact.

With the largest number of publications (97 publications), high volume of co-occurrence frequency (529), and co-occurrence centrality of 0.11, the *Adapted Physical Activity Quarterly* (APAQ) [57] exhibits a dominant position in the field of research on physical activity in disabled children and adolescents. The APAQ is an official journal of the *International Federation of Adapted Physical Activity* (IFAPA) [58]. The APAQ is a leading journal in multidisciplinary research on physical activity for individuals with disabilities and it spans disciplines such as physical education, human development, biomechanics, and psychology. The APAQ aims to enhance performance and participation in people with disabilities, covering adaptation in activities, environments, equipment, and rules across the lifespan.

*Disability and Health Journal* is the official journal of the American Association on Health and Disability (AAHD). Disability and Health Journal covers empirical research, systematic reviews, theoretical interpretations, and evaluations in disability and health, addressing global health, quality of life, and specific health conditions related to disability. The journal emphasizes issues, policies, and scientific approaches to enhance the health and well-being of individuals with disabilities [59].

The journal *Developmental Medicine and Child Neurology* (DMCN) [60] also holds a central position in this field and has a very high impact with a co-occurrence centrality of 0.16 and frequency of 539 (see Table 2). DMCN is a multidisciplinary journal in pediatric neurology and neurodisability, shaping these fields for over 60 years. Published by Mac Keith Press, it disseminates global clinical research to enhance care for disabled children and families, endorsed by various associations such as AACPDM, AusCPDM, BACD, etc.

Many of the top journals are affiliated with influential organizations such as IFAPA and AAHD. IFAPA is a global scientific organization that promotes adapted physical activity worldwide, coordinating activities in sports, dance, aquatics, exercise, and wellness for individuals with disabilities. AAHD ensures health equity and inclusion for individuals with disabilities through policy, research, and dissemination efforts [61]. Researchers can leverage these organizational partnerships to access resources, networks, and dissemination channels.

The influential journals in terms of co-occurrence (see Table 2) all have a very long publication history, with more than 50 years of publication experience, except for Research in Developmental Disabilities, Disability and Rehabilitation, and Adapted Physical Activity Quarterly. Two journals, Archives of Physical Medicine and Rehabilitation and Physical Therapy, have the longest publication history, with 104 and 103 years, respectively. Out of the top ten productive journals (see Table 1), only Developmental Medicine and Child Neurology and Journal of Intellectual Disability Research have a publication history of longer than 50 years. These journals provide a rich foundation of prior research to build upon. Researchers should thoroughly review this extensive body of work to situate their studies within the existing knowledge base.

By leveraging interdisciplinary collaboration, publishing in specialized journals, partnering with established organizations, building on decades of prior research, and addressing underexplored disability types, researchers can maximize the impact and advancement of the field of physical activity in disabled youth.

#### 4.1.2. Knowledge Structure in Terms of Countries

The United States (US) is the leading country in the research area of physical activity of disabled children and adolescents with a number of publications and a co-occurrence frequency of 495. The US has the largest research volume in the current research field, accounting for 38% of all publications. With the largest number of co-occurrence centrality (0.26), the US is also the most active country in inter-country collaborations. The US is the top country in terms of economy, technology, and research. According to the World Bank data, the gross domestic product (GDP) of the US was worth USD 25439.70 billion in 2022 [62]. The total research and development spending of the US was USD 789 billion in 2021, accounting for around 3.5% of the GDP.

The top four leading countries, the US, Canada, Australia, and the United Kingdom are all major developed economies (G7). Except for the People’s Republic of China, the top ten countries are all developed economies.

Considering the year of the first publication, Canada and the US emerged as pioneers, publishing related papers in 1995 [63] and 1996 [64], respectively. The Canadian pioneers identified and recognized that incorporating social skills training into structured physical activity programs can effectively improve both real motor skills and self-perceived competence in physical and academic domains [63]. The pioneer researchers in the US investigated the energy expenditure of children and adolescents with spastic quadriplegic cerebral palsy [64].

China, ranked the fifth position with 71 publications, is the leading developing country in studying the physical activity of disabled youth. China’s first publication investigated sports participation among Chinese school-aged children with disabilities in Hong Kong in 2002 [65]. The Chinese researcher found that disability type appeared more significant than gender or school level in shaping children’s participation behaviors in sports and physical activities [65].

Ireland, Saudi Arabia, and Romania published their initial papers in 2014 [66], 2016 [67], and 2020 [68], signifying a recent emphasis on research related to physical activities for children and adolescents with disabilities. This reflects the development of high-level academic teams and an expanding influence in this field.

#### 4.1.3. Knowledge Structure in Terms of Institutions

The leading institution studying the physical activity of disabled children and adolescents is the *University of Toronto*. The University of Toronto was founded in 1827 and evolved as Canada’s leading institution. The University shows a dominant position with the largest number of publications (75) and co-occurrence frequency (74) as well as large co-occurrence centrality (0.12), which indicates that the University of Toronto is at top in terms of both research volume and inter-institution collaborations.

At the University of Toronto, the Faculty of Kinesiology and Physical Education, Dalla Lana School of Public Health, and Temerty Faculty of Medicine and Rehabilitation Sciences Institute contributed most of the publications. These faculties are leading academic institutions in the studies of human movement, health, and physical activity, public health policies and practices, medical education, research, healthcare innovation, and rehabilitation.

### 4.2. Foundation of Research of Physical Activity in Children and Adolescents with Disability

We identified the fifteen most influential papers with significant reference citations, co-citation frequency, and citation burst strength, establishing the foundation of research in the field of physical activity in children and adolescents with disabilities. The papers cover a broad range of themes, including physical activity guidelines [11], the current status of physical activities in disabled youth [5,10,11,12,22,23,24], the influential factors, and perceived barriers and facilitators [25,26,27,28,29,30,45] aimed at enhancing physical activity levels among disabled youth [8].

#### 4.2.1. Physical Activity Guidelines for Children and Adolescents with Disabilities

In December 2020, the *British Journal of Sports Medicine* published a paper titled “World Health Organization 2020 Guidelines on Physical Activity and Sedentary Behaviour” [3]. The updated *World Health Organization* (WHO) guideline strongly advocates that children and adolescents engage in a minimum of 60 min per day of moderate-to-vigorous intensity aerobic physical activity throughout the week, incorporating vigorous-intensity aerobic activity, muscle-strengthening, and bone-strengthening activities at least three days a week. It is recommended that children and adolescents with disabilities should strive to meet these guidelines when feasible [3,4]. This recommendation aligns with existing research on school-age youth with disabilities [5], guidelines from the *American College of Sports Medicine* (ACSM) [69], and the *Canadian Society for Exercise Physiology* (CSEP) [70].

On the contrary, prolonged sedentary behavior, defined as any waking behavior with an expenditure ≤ 1.5 metabolic equivalents [71] while in a sitting, reclining, or lying posture, is associated with detrimental effects on fitness, cardiometabolic health, adiposity, behavior, and sleep duration [3]. Therefore, the WHO 2020 guideline strongly advises limiting sedentary time, particularly recreational screen time, for all children and adolescents, including those with disabilities.

#### 4.2.2. Current Status of Physical Activity in Children and Adolescents with Disabilities

Globally, a concerning 81% of adolescents aged 11~17 years are reported to be insufficiently physically active [72]. This pervasive physical inactivity poses a severe threat to the overall health and well-being of the population, necessitating urgent and comprehensive efforts to implement effective policies and programs aimed at increasing physical activity levels, particularly among children and adolescents.

The rates of physical inactivity among youth with disabilities are notably higher than their non-disabled counterparts, highlighting significant disparities across genders, regions, and countries [5,8,9]. A study conducted in Canada found that only 29% of the children with disabilities considered themselves physically active [71]. Another study investigated 3010 US children aged 6 to 17 diagnosed with different types of disabilities, such as autism spectrum disorders, cerebral palsy, Down syndrome, developmental disability, and/or intellectual disability [11]. The finding revealed that only 19% of the individuals met the national physical activity requirement of 60 min [11]. According to a comparison study on 319 adolescents aged 11 to 16 with their 7020 non-disabled counterparts, the disabled youth had a 4.5 times higher rate of physical inactivity compared to their healthy peers [73].

For young people with cerebral palsy (CP), research indicates a substantial reduction (13~53%) in habitual physical activity levels among affected youth aged 5~18 compared to their typically developing peers [10]. These disparities also include a 30% lower adherence to recommended activity guidelines and twice the recommended sedentary time [10]. According to a population-based survey of children with CP born in Victoria, Australia in 1994 and 1995, CP children participated in a median of 26.5 activities, predominantly informal rather than formal, with overall low intensity [22]. It is also noticeable that CP children participate more in organized sports than their non-disabled peers, albeit with lower frequency [22].

Examining intellectual disability, Icelandic school children with mild-to-severe intellectual disability were found to be 40% less physically active and spend 9% more time in sedentary activities than their typically developed peers. Only 33% of the children with intellectual disability participated in sports for more than 2 h per week, compared to 76% of the typically developed children. No children with intellectual disability met the recommendation of 60 min of daily moderate-to-vigorous physical activity [12].

Children with autism spectrum disorders (ASDs) aged 3~11 years demonstrated a similar amount of daily time spent in moderate and vigorous activity compared to typically developing children, as measured by accelerometers (50 min/day vs. 57.1 min/day) [23]. However, children with ASD participated in fewer types (6.9 vs. 9.6) of physical activities and spent less time annually (158 vs. 225 h per year) engaging in these activities [23].

Furthermore, adolescents with physical disabilities encounter challenges in sports team participation [9], and they exhibit limited engagement in school-based physical activities [13]. This contributes to the higher levels of sedentary behaviors during weekends [13]. Physically disabled adolescents were reported twice as likely as their peers without a disability to engage in sedentary activities such as watching television for more than 4 h each day [5]. The proportion of students with disabilities who engaged for more than 3 h per school day in sedentary activities, such as video or computer games, was 26.5%, significantly longer than the non-disabled students (20.4%).

In conclusion, children and adolescents with disabilities display notable physical inactivity in terms of the insufficient duration of moderate-to-rigorous physical activity, fewer activity types, fewer formal sports, and more organized sports, although there are disparities across disability types, genders, regions, and countries.

The high rates of physical inactivity in youth with disability put them at a significantly greater risk of developing serious health problems, both physically [3,4] and psychologically [3,4,5]. Compared with their non-disabled peers, children or adolescents with disabilities are more likely to experience an increased risk of obesity, cardiovascular disease, type 2 diabetes, or other chronic conditions [29]. They may also have lower cardiorespiratory fitness and muscle strength, poorer bone health, and an increased risk of osteoporosis. Additionally, they are more likely to experience higher rates of depression, anxiety, and other mental health issues [3,4,5,29]. These long-term consequences can persist into adulthood, leading to a lifetime of poorer health outcomes and reduced independence for individuals with disabilities [3,4,5,29].

#### 4.2.3. Related Factors, and Perceived Barriers and Facilitators to Physical Activity Promotion in Children and Adolescents with Disabilities

To boost physical activity in disabled youth, a thorough understanding of key influencing factors is essential. Health professionals, teachers, policy makers, and sports clubs can utilize this insight to develop targeted interventions for increased participation. Out of the fifteen foundation papers, five papers, including both review papers and qualitative study papers, addressed the influential factors and perceived barriers and facilitators to the promotion of physical activity in children and adolescents with disabilities.

First of all, the term “participation” should be clarified. According to the *International Classification of Functioning, Disability, and Health for Children and Youth* (ICF-CY) [74] by the World Health Organization, “participation” stands out as the ultimate health outcome and serves as both the starting point and culmination of intervention. The ICF defines participation as “involvement in a life situation” and “participation restriction” as “problems individuals may encounter in involvement in life situations”. Put simply, “participation” involves attending and engaging in life situations, with “attendance” indicating presence measured by the frequency or diversity of activities, and “involvement” encompassing the experience of participation, including engagement, motivation, persistence, social connection, and affect [30,45].

Manon and colleagues published a systematic review in 2014, which summarized the important factors associated with participation in physical activity for children and adolescents with disabilities [27]. They classified all related factors into two categories, personal factors and environmental factors.

The personal factors encompass “intention”, “attitude”, “self-efficacy”, “health condition”, and “personal barriers and facilitators” [27]. Regarding “intention”, the facilitator is the “desired to be active”. At the level of “attitude”, there are several facilitators including a “positive thought toward an active life style”, a “desire to be healthy”, the recognition of the “importance and benefits of physical activity”, etc. Conversely, barriers at this level encompass the belief that “being active is not good for the body”, the need for “rest in spare time”, and the “fear of safety, injury or incontinence”. Concerning “self-efficacy”, barriers include “feeling insecure”, “lack of confidence”, and perceiving “an attractive sport is too difficult”. On the contrary, “feeling confident”, “gaining self-confident”, and “gaining sport competence” were positive factors. At the level of “health condition”, the level of “gross motor function classification system (GMFCS)” is significantly associated with physical activity. Exploring “personal barriers and facilitators”, factors related to the child’s “fitness”, “motivation”, and “abilities” are identified. Interestingly, the “increase of age” is markedly negatively associated with participation in physical activities [51,75].

The environmental factors encompass “social influence” and environmental “barriers and facilitators” [27]. Regarding “social influence”, parental influence emerges as the sole facilitator. The belief that “physical activity is important for the child” plays a pivotal role in promoting their child’s physical activity. Concerning “environmental barriers and facilitators”, various physical and social environmental factors were identified. “Family, teacher, peers, and other people” can either impede or encourage physical activity. The factor of “supporting from teachers and instructors” is critical and positively correlates with physical activity [76]. “Adequately adapted activities, programmes and equipment” and “access to transport” were constantly reported as positive factors when available, and conversely, as negative factors when absent. It is noteworthy that negative factors in this domain were also associated with sport facilities [26].

Ginis and her colleagues published a systematic review of review articles addressing factors influencing physical activity participation among individuals with physical disabilities in 2016 [28]. This impactful review has been cited 261 times within the core collection of the WoS dataset (see Table 6). Kathllen et al. extracted over 200 factors from 22 review articles, focusing on barriers and facilitators to leisure time physical activity (LTPA) in both children and adults with physical disabilities. Utilizing a social–ecological model, the 208 factors were classified into five levels: the intrapersonal level, the interpersonal level, the institutional level, the community level, and the policy level [28].

At the intrapersonal level, four themes were identified, including “psychological factors”, “body functions and structures”, and “employment status” [28]. Key elements within psychological factors encompassed “affect and emotion”, “attitudes/beliefs/perceived benefits” and “self-perceptions”. Notably, negative mood, depression, anxieties, fears, and embarrassment related to activity were commonly identified as affective/emotional barriers. Conversely, positive attitudes and beliefs about being active (e.g., providing opportunities to meet others and enhancing function) and about oneself (e.g., self-efficacy and self-determination) were frequently acknowledged as facilitators.

Regarding the interpersonal level, factors were categorized into three themes: “social support”, “attitudes”, and “social processes” [28]. The reviews consistently highlighted the significance of support from family, friends, peers, healthcare professionals, and other professionals in facilitating physical activity (all ICF-labeled sub-themes). Conversely, negative attitudes from others (also an ICF label) were frequently identified as a hindrance to activity. Role modeling was frequently cited as positively associated with sports and exercise participation.

In terms of the institutional level, the factors were “knowledge within institutions/organizations”, “rehabilitation processes”, “building design and construction” (an ICF label), and “program factors” [28]. Knowledge levels in healthcare professionals and service providers significantly influenced participation. Disability-specific knowledge about the benefits of physical activity and exercise was consistently highlighted. Rehabilitation processes emphasized the importance of information, counseling, and encouragement from professionals. Building accessibility and location were crucial factors. Program-related factors, such as “availability” and “fun/enjoyable activities”, are positively associated with physical activity participation.

At the community level, factors revealed the themes of “products and technology”, “climate”, and “relationships among groups and organizations” [28]. These align with McLeroy et al.’s inclusive definition of “community”, encompassing structures and group relationships. Within “products and technology”, three sub-themes with ICF labels are identified: “product and technology for culture, recreation and sport”, “land development”, and “education”. Equipment and information played crucial roles. Climate influenced LTPA participation. Four factors related to “relationships among groups and organizations” were identified but not frequently cited overall.

At the policy level, five themes were identified [28]. Two ICF-labeled themes, health policies (specifically, funding for programs) and transportation services/systems/policies, were frequently cited within government policy purview. Association and organizational policies were crucial, with financial costs and staff training identified as common barriers.

### 4.3. Research Hotspots

Based on the keyword co-occurrence analysis, clustering, and burst analysis, three research hotspots are identified: skill, competence, and scale.

#### 4.3.1. Skill

Training or interventions aimed to enhance skills, including both motor [77,78,79,80,81,82,83,84] and cognitive skills [84,85,86], are proposed for disabled young people.

Tailored cycle training [78] and manual wheelchair (MWC) skill training [82] demonstrated positive outcomes, increasing the intention to cycle and enhancing MWC skill capacity, self-efficacy, and satisfaction with participation. Michelle et al.’s focus group interviews [77] identified training needs for supporting students with visual impairments, aiming to improve the motor skills of students with visual impairments. Virtual reality (VR)-assisted exergaming therapy [84] significantly improved outcomes for children with cerebral palsy, enhancing various motor and cognitive dimensions [84]. Aslan et al.’s study [81] highlighted the effectiveness of computer-aided video modeling in teaching basic basketball movements to individuals with Down syndrome, leading to an increased self-confidence and strengthened peer relations through active participation.

Physical activity interventions also play a crucial role in enhancing communication and social interaction for individuals with autism spectrum disorders [85]. Specifically, a football program demonstrated notable improvements in overall psychosocial skills, encompassing verbal communication, social interaction, transition, and task attention, among youth with autism spectrum disorder [86]. Additionally, a school-based running program proved effective in enhancing fitness and the quality of life for children with physical and cognitive disabilities [83]. Furthermore, a combined visual arts and exercise program exhibits improvements in communication skills and social behavior for students with autism spectrum disorders [87].

#### 4.3.2. Competence

Competence is one of the main concepts in the *self-determination theory* (SDT) framework [88]. Competence pertains to an individual’s subjective sense of capabilities and effectiveness in achieving desired outcomes and interacting with the environment [89]. It influences engagement in challenging tasks, serving as a source of energy in behavior and learning processes, emphasizing subjective mastery over objective abilities. Another two important factors in the SDT framework are the satisfaction of autonomy and relatedness, which are human’s basic needs in combination with competence. In the SDT framework, an individual’s perceptions of need fulfillment can facilitate the transition from a more controlled motivation to a more autonomous one. The autonomous motivation for physical activity is associated with a greater engagement and persistence in activities.

There are strong positive correlations between the satisfaction of autonomy, competence, and relatedness and the students’ autonomous motivation, which is positively related to performance, athlete’s well-being, and sport adherence. For self-organized physical activity and exercise, autonomous motivation, facilitated by the means of psychological need fulfillment, is the most prominent factor for participation and adherence to physical activity across diverse samples and contexts [90,91].

Marte et al. assessed 4140 Norwegian students aged 13~19 years attending secondary or upper secondary school [88]. Those students included 328 individuals with disability/long-term illness. They measured the physical activity level and the perceived fulfillment of the needs for competence, autonomy, and relatedness using self-developed questionnaires and the Norwegian adaptation of the Basic Psychological Needs in Exercise Scale (BPNES) [92] at three time points. Adolescents with disability/long-term illness reported significantly lower autonomy, competence, and relatedness in physical education compared with their peers in all three time points. In organized sports, although adolescents with disability/long-term illness scored lower for autonomy, competence, and relatedness, these differences diminished in T2 and T3. For self-organized physical activity, adolescents with disability/long-term illness had significantly lower autonomy compared with those without. The findings demonstrate that need fulfillment is varied across different physical activity contexts, suggesting that the satisfaction of basic needs is both dynamic and context-dependent. This study indicates that young people with disability are vulnerable to experiencing a reduced participation in physical education and organized sports. It is recommended that rehabilitation institutes, physical education teachers, and coaches should work closer to support adolescents’ three psychological needs in physical activity regardless of disability status and give them experiences with meaningful participation in physical activity [88]

Daniel and colleagues developed a participatory action project, named *Game Changer*, aiming to enhance inclusive school sport opportunities for students with disabilities [93]. In the pilot study, twenty-seven students with various cognitive and/or intellectual disabilities (student-participants), their six school physical education teachers and learning support teachers (champion-participants), four university researchers (researcher-participants), and two community partners from *Physical and Health Education* (PHE) Canada, Special Olympics Nova Scotia collaborated in the Game Changer project. The project’s three idealized goals are as follows: “(a) highlighting para/adapted/inclusive sport opportunities for all students; (b) empowering opportunity for students with disabilities to participate, make choices, and act as leaders in the development of sport programming; and (c) engaging youth with disabilities in sport as participants, leaders, mentors, and role models”.

The Game Changer project consists of six stages, requiring collaboration among participants and partners over a period of under one year. It followed a cyclical process of “actioned” research with five sequential and recurrent stages: think, plan, act, evaluate, and reflect. Following the first cycle of the Game Changer project, several positive outcomes were evident, such as enhancing students’ perceived competence and autonomy, encouraging student voice, identifying and responding to sport participation barriers, and establishing genuine sport opportunities in school settings [93].

#### 4.3.3. Scale

In recent years, researchers employed or developed numbers of scales to quantify almost all research areas in the physical activities of disabled youth, including both physical measures such as the levels of physical activity as well as body functions and psychological measures.

In terms of the physical measures, the *International Physical Activity Scale-Short Form* (IPAQ-SF) was employed to assess the physical activity level of children with physical disabilities and typically developing children [94], enabling a comparison of the physical activity levels between groups. The IPAQ-SF was proved to have a high reliability (ranging from 0.66 to 0.88) [95,96,97]. However, the validity of IPAQ-SF has shown varied results across different studies. According to a systematic review [98], the correlation between total physical activity levels assessed by the IPAQ-SF and objective golden standards fell below the minimal acceptable standard in the literature (0.50 for objective activity measuring devices and 0.4 for fitness measures), with the correlation value of 0.09 to 0.39. On average, the IPAQ-SF overestimated physical activity compared to objective measures by 84 percent [98]. The validity of IPAQ-SF in assessing moderate-to-vigorous physical activity (MVAP) shows a greater variability across studies (−0.18 to 0.76), yet several reached the minimally acceptable standard [98]. A recent meta-analysis found that IPAQ-SF is best to measure the MVPA, with no publication bias, moderately high test–retest reliability (0.74), and moderately high concurrent validity (0.72) in adults across EU countries [99].

The *Pediatric Balance Scale* (PBS), quantifying balance during movement, was used to study the effect of strengthening and aerobic exercises on balance and functional independence levels in young people with various neurological and musculoskeletal conditions [100]. Studies show that PBS is both valid [101] and reliable [102,103]. The PBS has a moderate-to-high correlation (0.448~0.579) with clinical balance assessments such as the balance measure when eyes are open and closed and fixed foot support, gross motor function measure, and the pediatric evaluation of disability inventory mobility skill [101]. Furthermore, the PBS score can distinguish between different gross motor function Classification Scale (GMFCS) levels in children with cerebral palsy [101]. The reliability of PBS is high with the intraclass coefficients (ICCs) of 0.89~0.99, including the PBS in different languages such as Greek and Turkish [102,103].

*Functional Mobility Scale* (FMS) was used to quantify physical activity levels in ambulant/semi-ambulant children and adolescents with cerebral palsy [100]. The FMS demonstrates good validity and reliability [104,105]. Specifically, the FMS has excellent inter-rater reliability with ICCs of 0.92~0.99 [104] and a very strong correlation with the GMFCS (−0.85 < r < −0.89) [105]. The FMS in other languages such as Greek also demonstrated good reliability [105].

*Early Activity Scale for Endurance* (EASE) was developed to evaluate the endurance for physical activity of preschool children with *cerebral palsy* (CP) [106]. The EASE differs significantly from GMFCS, correlated moderately (0.57) with the 6 min test and the ICCs between the two tests was 0.95, showing good validity and reliability in young children with CP [106]. The Turkish version of the EASE (T-EASE) was also found to be valid and reliable [107] in estimating the endurance for physical activity of Turkish preschool children with CP.

The *Contour Drawing Rating Scale* (CRDS), the *Child Feeding Questionnaire* (CFQ), and the *Emotional Overeating Questionnaire* (EOQ) were used in the study to assess the relationship between engaging in physical activity and emotional eating in children and adolescents with mild and moderate intellectual disability [108]. The CRDS was found to be highly correlated (0.64) with measured BMI and exhibited a high test–retest reliability for current size, ideal size, and current–ideal discrepancy (0.65~0.87) [109]. The CFQ was proved to have adequate reliability and validity with the Cronbach’s alpha coefficient of 0.85 and 0.78 in restriction and pressure to eat [110]. The EOQ was used to evaluate the frequency of overeating and demonstrated adequate reliability and validity with the Cronbach’s coefficient of 0.97 [111].

To investigate the effect of disability type on perceived physical activity constraints, the *Leisure Time Physical Activity Constraints Scale* (LTPACS)*-Disabled Individuals Form* was used for data collection [112]. Using the LTPACS, researchers found that people with different disabilities have different perceived leisure time physical activity barriers [112]. However, the validity and reliability of the LTPACS were not available [112].

Regarding the psychological measures, the *Expanded Disability Status Scale* (EDSS) was used to quantify and monitor changes in disability of children and adolescents with multiple sclerosis (MS) [113]. The EDSS is considered valid in assessing disability and disease progression in patients with MS, with strong correlations between EDSS scores and the other measures of disability and function such as the Barthel Index (−0.74) [114]. The EDSS also demonstrated adequate discriminant validity with its ability in distinguishing the different levels of disability [115]. But the reliability, especially inter-rater reliability has been questioned [115].

The *Chedoke-McMaster Attitudes towards Children with Handicaps Scale* (CATCH) could be used to measure the attitudes of children toward peers with disabilities [116]. The CATCH was proved to be valid and reliable with the Kaiser–Meyer–Olkin values of greater than 0.90 and the Cronbach’s alpha value of 0.85 [117]. A modified version of CATCH developed specifically for traditional sporting games (CATCH-TSG) was approved as a valid and reliable tool in the assessment of the change in attitudes toward people with disabilities after receiving a psychoeducational intervention that includes physical activity [118].

In order to investigate the association of physical activity with psychological distress and happiness in the mothers of children with autism spectrum disorders, the *six-item Kessley Psychological Distress Scale* (K6) and the *Subjective Happiness Scale* (SHS) were employed, enabling quantitive analysis between these two factors [119]. K6 has demonstrated good validity for assessing psychological distress across various settings and populations with high correlations with Generalized Anxiety Disorder (0.66) and Somatic Symptoms Scale (0.61) and high factorial correlations (0.60) [120]. K6 also demonstrated an excellent internal reliability with Cronbach’s alpha coefficients ranging from 0.89 to 0.92 and good test–retest reliability with the ICCs from 0.77 to 0.89 [120]. Current evidence indicates that the SHS is a valid and reliable instrument for measuring subjective happiness across diverse populations and cultural contexts, correlating strongly with the other measures of happiness and well-being, such as the Affect Balance Scale and the Satisfaction with Life Scale, with Cronbach’s alpha values of 0.85~0.92 and good test–retest reliability (0.55~0.90) [121,122,123].

*Rosenberg Self-Esteem Scale* (RSES) was used to quantify the relationship between physical activity, educational attainment, and the use of mobility aids with self-esteem in people with physical disability [124]. The RSES has demonstrate good validity with other measures of self-esteem or self-worth [125,126,127]. Moreover, the RSES has also shown excellent internal consistency with Cronbach’s alpha value of 0.83 to 0.92 and good test–retest reliability (0.55~0.90) [125,126,127].

### 4.4. Future Research Directions

Based on the interpretation of the current knowledge structures, research foundations, and research hotspots, the following future research directions are proposed:

Firstly, advocate and support global research on physical activity in children and adolescents with disabilities, especially in developing and underdeveloped countries, to improve the well-being of disabled youth and therefore the entire society. Moreover, encourage collaboration between countries, especially between developed countries and those that are not.

Secondly, conduct systematic research encompassing broader perspectives, including intrapersonal factors (e.g., psychological factors, body functions, etc.), interpersonal factors (e.g., social support, attitudes, social processes, etc.), institutional factors (e.g., disabled-related knowledge, rehabilitation process, building design, construction, etc.), community factors (e.g., product and technology, the relationship among groups and organization, etc.) and policy factors (e.g., health policy, transportation system/policies, etc.). Emphasize evidence-based quantitative studies on the above-mentioned aspects to promote the physical activity participation of youth with disabilities.

Thirdly, establish a practical guide by incorporating the policy makers, healthcare professionals and behavior scientists, physical stakeholders (e.g., the physical education teachers or coaches), family members or peers, and the disabled child or adolescent him/herself, to promote the life-long, optimal, inclusive physical activity participation of disabled youth.

Lastly, develop and implement personalized intervention for each child or adolescent with disabilities, utilizing emerging technologies such as information technology, intelligent sensors and equipment, tele-medicine systems, etc., in physical-activity-promotion practice.

### 4.5. Limitations

There are mainly two limitations which should be addressed. First, the current bibliometric review is based on the analysis of relative data extracted solely from the core collection of the Web of Science. Related publications from other datasets will be included for further analysis in the future; second, this review only analyzed data from January 1995 to December 2023, and therefore, the knowledge structure, research foundation, hotspots, and future directions only represent research in the 1995~2023 period. This knowledge structure should be continually updated in the future.

## 5. Conclusions

We conducted the first comprehensive bibliometric review on research related to physical activity in children and adolescents with disabilities, analyzing the related 1308 papers from the core collection of the Web of Science database using CiteSpace. The knowledge structure, including annual publication numbers, influencing journals, countries, institutions, authors, references, and keywords along with their respective collaborative networks, has been constructed. Furthermore, the research foundation, current hot topics, and research frontiers have been identified by analyzing references and keywords.

The number of publications showed a substantial increase from 1995 to 2023, particularly after the COVID-19 pandemic. Previous research has laid the foundation for the field of the research area of physical activity among disabled youth, providing insights into the guidelines, current status, influencing factors, and perceived barriers and facilitators. Current research hotspots include interventions, therapies, and programs aimed at enhancing specific skills, as well as addressing the satisfaction of competence to improve motivation and the effectiveness of physical activity. There is also a focus on the development of scales for quantitative studies.

In the future, establishing a practical guide by integrating health policy makers, healthcare professionals, scientists, physical education teachers or coaches, family members, peers, and disabled individuals and developing and implementing personalized interventions or programs using advanced technologies has the potential to enhance physical activity levels among youth with disabilities.

## Figures and Tables

**Figure 1 healthcare-12-00934-f001:**
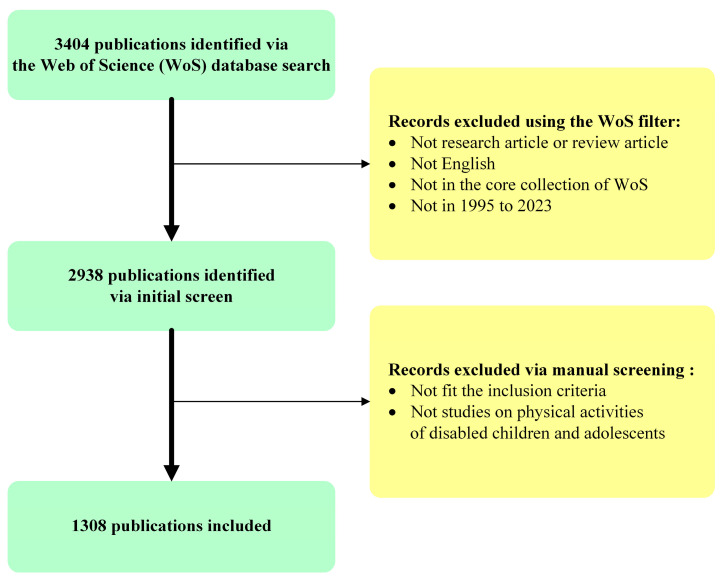
The literature screening flowchart.

**Figure 2 healthcare-12-00934-f002:**
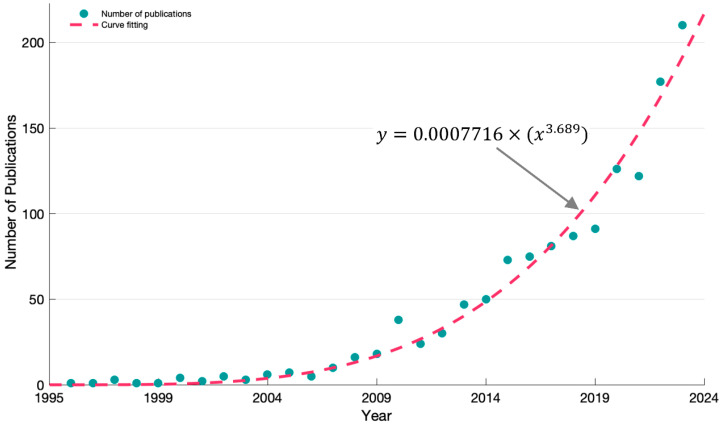
Annual number of published articles from 1995 to 2023. The green dot is the number of publications per year. The red dashed line is the fitted curve using the power function.

**Figure 3 healthcare-12-00934-f003:**
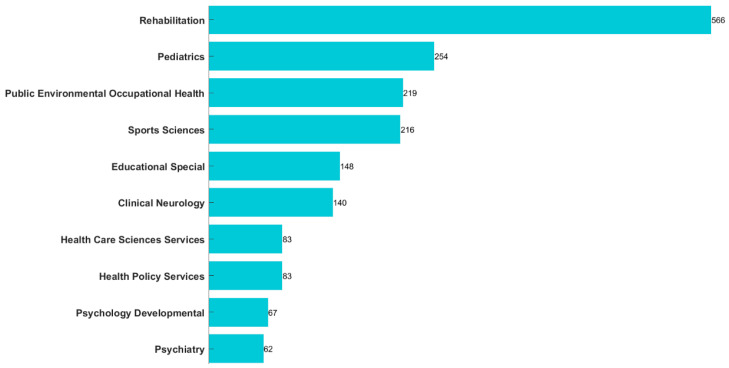
The Web of Science categories of all journals that published research on physical activity in disabled children and adolescents.

**Figure 4 healthcare-12-00934-f004:**
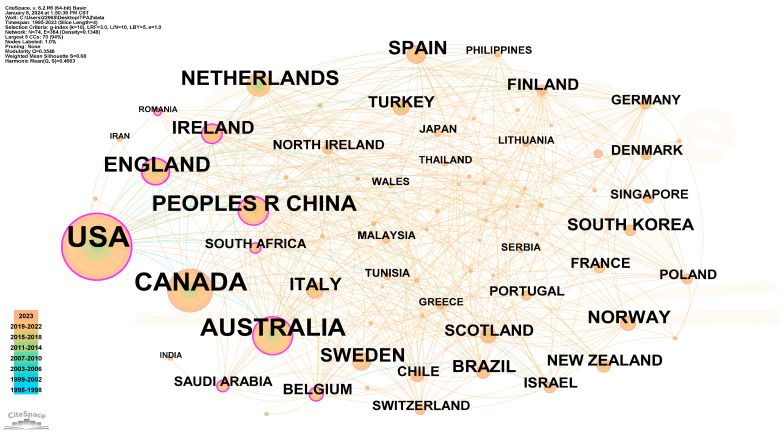
Co-occurrence map of countries conducting research on the physical activity of disabled children and adolescents. The size of the node circle in the figure corresponds to the volume of the related literature published by each country. The timeline, indicated in the lower left corner, showcases the publication year distribution, with earlier years represented by blue and later years by orange.

**Figure 5 healthcare-12-00934-f005:**
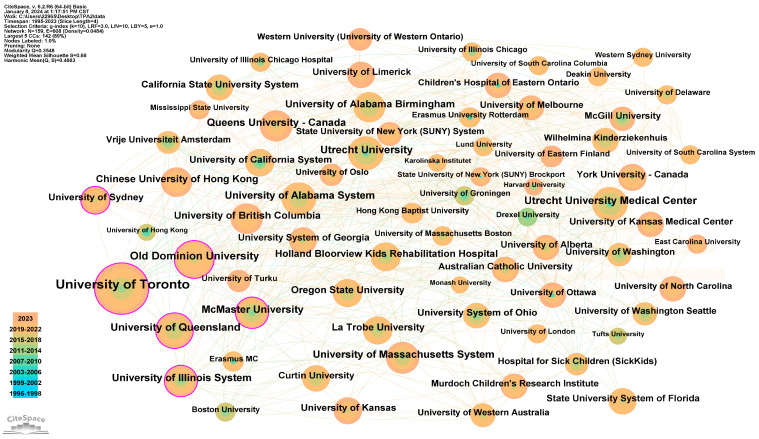
Institution collaboration map of physical activity among children and adolescents with disabilities (19952023). Node size represents the volume of the related literature published, and the timeline is displayed in the lower left section.

**Figure 6 healthcare-12-00934-f006:**
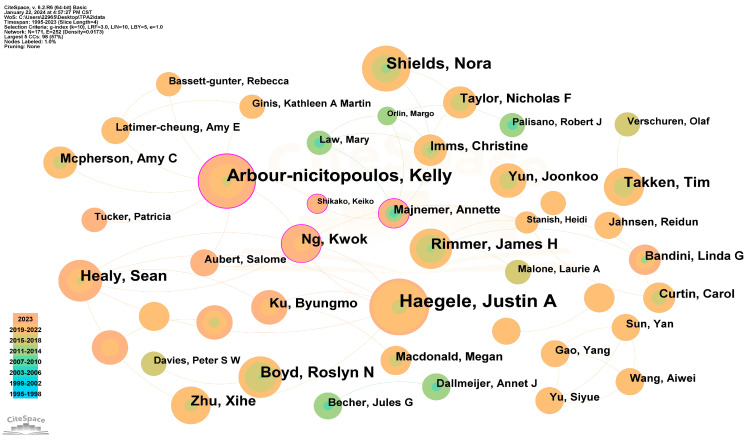
Author cooperative network map. The size and color of each circle represent the year of the author’s publication, indicated by the color band in the lower-left corner corresponding to the year. Collaboration between authors is represented by linked chains.

**Figure 7 healthcare-12-00934-f007:**
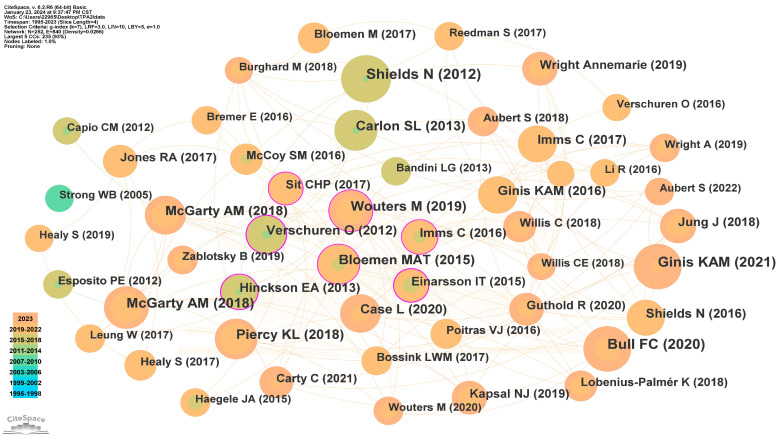
The co-citation reference network. The size and color of each circle represent the year of the reference’s publication, indicated by the color band in the lower-left corner corresponding to the year.

**Figure 8 healthcare-12-00934-f008:**
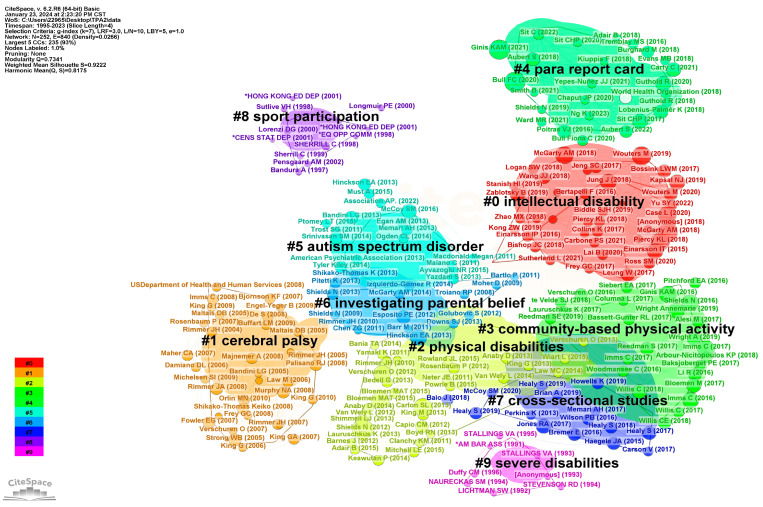
The clustering diagram of the references. The lable* represents literature with high citation frequency or impact.

**Figure 9 healthcare-12-00934-f009:**
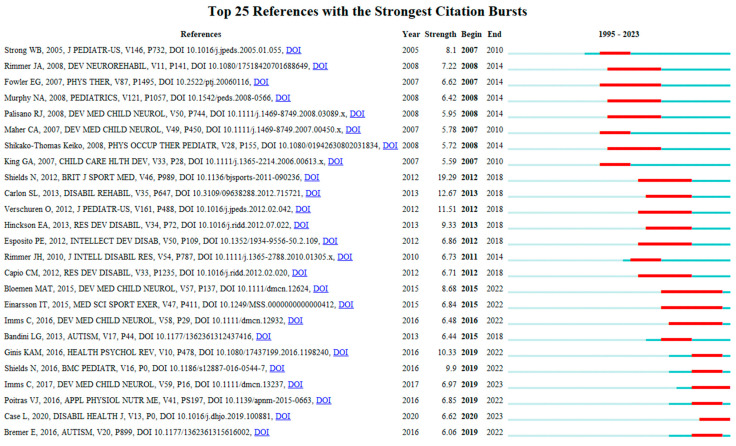
Top references with the strongest citation bursts [5,8,10,11,12,23,24,25,26,27,28,29,30,45,46,47,48,49,50,51,52,53,54,55,56].

**Figure 10 healthcare-12-00934-f010:**
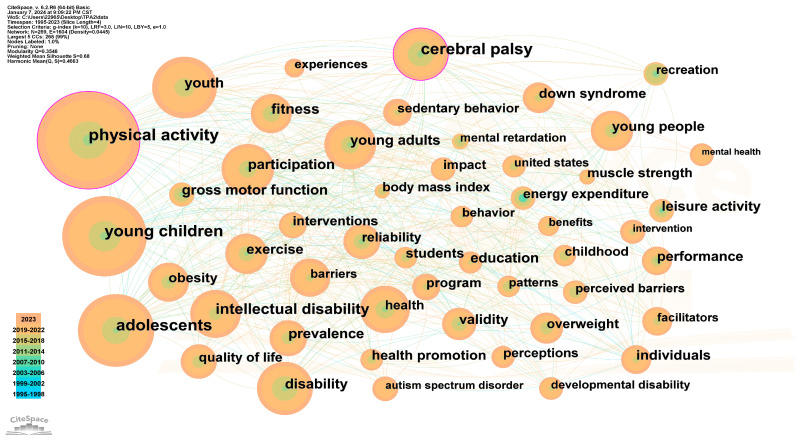
The co-citation map of the keywords.

**Figure 11 healthcare-12-00934-f011:**
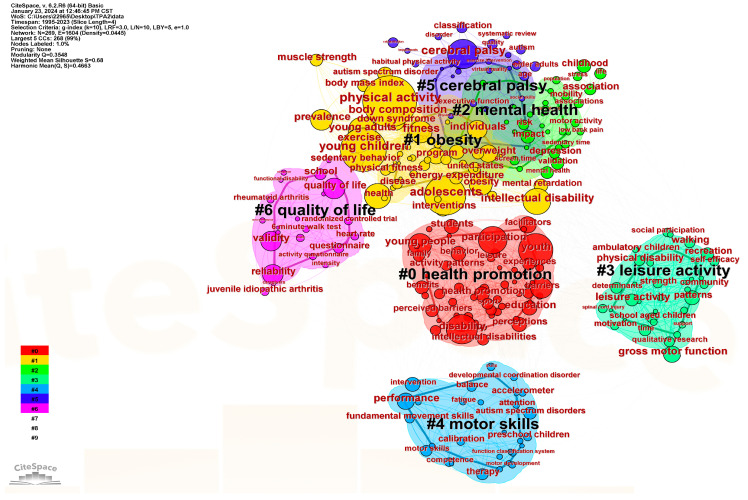
The clustering analysis map of the keywords.

**Figure 12 healthcare-12-00934-f012:**
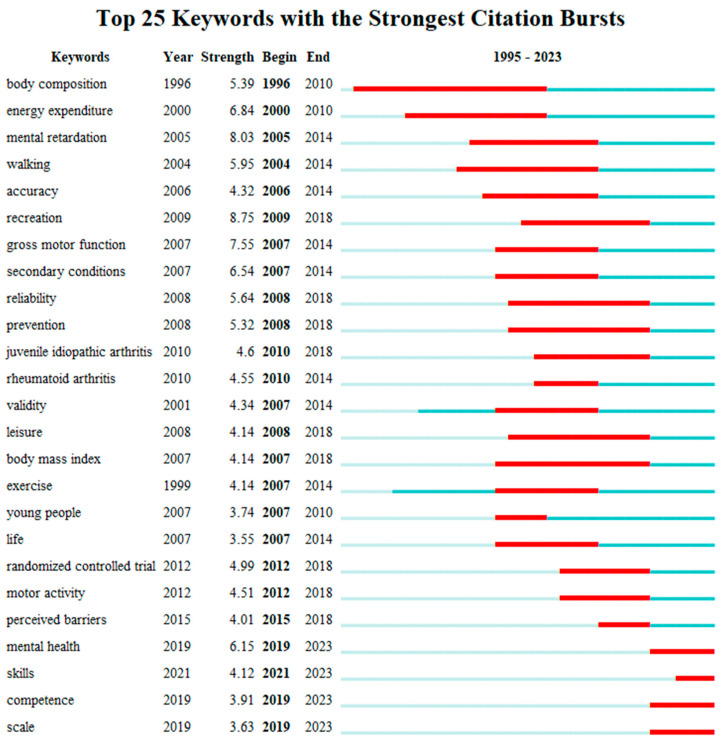
Top 25 keywords with the strongest citation bursts.

**Table 1 healthcare-12-00934-t001:** Top 10 most productive journals related to physical activities of disabled children and adolescents.

Rank	Journals	Impact Factor *	Publications	Countries
1	Adapted Physical Activity Quarterly	1.9 ^a^	97	United States
2	Disability and Health Journal	4.5 ^a,b^	62	United States
3	Disability and Rehabilitation	2.2 ^a,b^	60	England
4	International Journal of Environmental Research and Public Health	4.614 ^a,b,#^	46	Switzerland
5	Developmental Medicine and Child Neurology	3.8 ^a^	40	England
6	Research in Developmental Disabilities	3.1 ^b^	34	United States
7	Journal of Intellectual Disability Research	3.6 ^b^	28	England
8	Pediatric Physical Therapy	1.6 ^a^	27	England
9	BMJ Open	2.9 ^a^	24	United States
10	Journal of Applied Research in Intellectual Disability	2.4 ^b^	24	England

* According to Journal Citation Reports (2023); ^#^ according to Journal Citation Reports (2022); ^a^ Science Citation Index Expanded; ^b^ Social Science Citation Index (SSCI).

**Table 2 healthcare-12-00934-t002:** Top 10 most frequently co-occurred journals related to physical activities of disabled children and adolescents.

Rank	Cited Journals	Impact Factor *	Centrality	Frequency	Countries
1	Research in Developmental Disabilities	3.1 ^b^	0.06	625	United States
2	Medicine and Science in Sports and Exercise	4.1 ^a^	0.09	564	United States
3	Disability and Rehabilitation	2.2 ^a,b^	0.14	553	England
4	Developmental Medicine and Child Neurology	3.8 ^a^	0.16	539	England
5	Adapted Physical Activity Quarterly	1.9 ^a^	0.11	529	United States
6	Pediatrics	8.0 ^a^	0.10	449	United States
7	Archives of Physical Medicine and Rehabilitation	4.3 ^a^	0.05	397	United States
8	Physical Therapy	3.2 ^a^	0.04	372	United States
9	Journal of Intellectual Disability Research	3.6 ^b^	0.08	347	England
10	British Journal of Sports Medicine	18.4 ^a^	0.04	330	England

* According to Journal Citation Reports (2023); ^a^ Science Citation Index Expanded; ^b^ Social Science Citation Index (SSCI).

**Table 3 healthcare-12-00934-t003:** Ranking of the top ten countries on the physical activity of disabled children and adolescents with the number of co-occurrence frequency, co-occurrence centrality, and the year of first publication.

Rank	Frequency	Countries	Year	Centrality
1	495	United States	1996	0.26
2	217	Canada	1995	0.05
3	170	Australia	2000	0.13
4	82	England	2003	0.22
5	71	People’s Republic of China	2002	0.14
6	69	Netherlands	2009	0.02
7	50	Spain	2006	0.05
8	46	Sweden	2001	0.01
9	41	Ireland	2014	0.13
10	38	Italy	2006	0.08

**Table 4 healthcare-12-00934-t004:** Top ten institutions in terms of the number of co-occurrence frequency on the research of physical activity among children and adolescents with disabilities. The co-occurrence centrality and the year of first publication are also demonstrated.

Rank	Frequency	Institution	Centrality	Year
1	74	University of Toronto	0.12	2007
2	35	Old Dominion University	0.15	2015
3	33	University of Queensland	0.10	2011
3	33	University of Alabama System	0.01	2012
5	32	Utrecht University	0.01	2009
5	32	Utrecht University Medical Center	0.01	2009
7	30	University of British Columbia	0.01	2007
7	30	University of Massachusetts System	0.07	2008
9	29	McMaster University	0.15	2007
10	29	University of Illinois System	0.11	2007
10	29	Queens University—Canada	0.02	2015
10	29	Oregon State University	0.09	2009
10	29	University of Alabama Birmingham	0.01	2012

**Table 5 healthcare-12-00934-t005:** Top five frequency and centrality of the cited authors of the collaborative network related to physical activity among disabled children and adolescents.

Rank	Frequency	Author	Rank	Centrality	Author
1	29	Haegele, Justin A	1	0.22	Arbour-nicitopoulos, Kelly
2	25	Arbour-nicitopoulos, Kelly	2	0.17	Ng, Kwok
3	18	Shields, Nora	3	0.16	Majnemer, Annette
4	15	Healy, Sean	4	0.16	Shikako, Keiko
5	15	Boyd, Roslyn N	5	0.07	Stanish, Heidi

**Table 6 healthcare-12-00934-t006:** The top 15 important papers with high citation count, reference burst, and centrality.

Rank	Title	Citation Count	Burst	Centrality
1	Perceived barriers and facilitators to physical activity for children with disability: a systematic review [25]	298	19.23	0.06
2	Participation, both a means and an end: a conceptual analysis of processes and outcomes in childhood disability [30]	324	6.97	0.01
3	Promoting the participation of children with disabilities in sports, recreation, and physical activities [8]	278	6.42	0.03
4	A systematic review of review articles addressing factors related to physical activity participation among children and adults with physical disabilities [28]	261	10.33	0.03
5	‘Participation’: a systematic review of language, definitions, and constructs used in intervention research with children with disabilities [45]	236	6.48	0.13
6	Differences in habitual physical activity levels of young people with cerebral palsy and their typically developing peers: a systematic review [10]	216	12.67	0.06
7	Perceived barriers and facilitators to participation in physical activity for children with disability: a qualitative study [29]	216	9.90	0.04
8	Physical activity for youth with disabilities: A critical need in an underserved population [5]	208	7.22	0.02
9	Comparison of physical activity between children with autism spectrum disorders and typically developing children [23]	177	6.44	0.02
10	Identification of Facilitators and Barriers to Physical Activity in Children and Adolescents with Cerebral Palsy [26]	157	11.51	0.23
11	Measuring physical activity in children and youth living with intellectual disabilities: a systematic review [24]	133	9.33	0.23
12	Diversity of participation in children with cerebral palsy [22]	133	5.35	0.13
13	Factors associated with physical activity in children and adolescents with a physical disability: a systematic review [27]	116	8.68	0.19
14	Differences in physical activity among youth with and without intellectual disability [12]	80	6.83	0.29
15	Physical activity guideline compliance among a national sample of children with various developmental disabilities [11]	52	6.62	0.05

## Data Availability

The original contributions presented in the study are included in the article, further inquiries can be directed to the corresponding authors.

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
