# Peer review of "Exploring Physical Activity in Children and Adolescents with Disabilities: A Bibliometric Review of Current Status, Guidelines, Perceived Barriers, and Facilitators and Future Directions"

_healthcare, 2024, doi:10.3390/healthcare12090934_

Round 1

Reviewer 1 Report

Comments and Suggestions for Authors

The issues discussed in the article are interesting, but the work requires some corrections and additions.

Its additional advantage is its carefully designed graphic design.

The purpose of the research is defined too generally - it should be made more specific.

I would suggest adding the names of the countries where the above-mentioned magazines are published in tables 1 and 2.

It would also be worth listing the magazines with the longest publishing experience. This is very important information.

It would be interesting to analyze in relation to continents.

I would suggest showing how the percentage of disabled people changes in the analyzed period of 1995-2023. This could perfectly complement the information in Figure 2.

Reviewer 2 Report

Comments and Suggestions for Authors

General Comments

I do like to start by thanking you for the opportunity to review this manuscript.

The manuscript addresses a very interesting and pertinent topic, frames it well and justifies its relevance. The methodology used is adequate but has some limitations that should be clearly addressed and, above all, justified (e.g. use of only one database, time limitation of the research). The results are interesting and will contribute to greater knowledge on the subject, but the discussion needs to be deeply improved in some sections which are similar to the presentation of results.

If the authors address all of the suggestions for improvement, the manuscript could be considered for publication. Below are my suggestions by section and with the identification of the lines.

 Introduction

Lines 42-45 – Please add a reference to support this sentence.

Lines 49-56 - There are several theoretical models in the areas of motor development and behavior that support all these statements, it would be interesting to explore them a little, or at least cite them to reinforce this paragraph. Suggestions: (Hulteen et al., 2018; Stodden et al., 2008).

General comment for the introduction - Fluid and well-structured introduction. It is advisable to reinforce with more references on the points identified above.

 Material and Methods

Line 101 – Please justify why the authors only included one database.

Lines 110-111 – Please explain why the authors only considered articles after 1995, couldn't earlier articles have enriched the research?

Figure 1 - I understand that this review is a Bibliometric review and not a systematic review following the PRISMA protocol. Nevertheless, there are some points from the prism flow chart that could be applied here and enrich it. I suggest including in the 2nd yellow rectangle the list of reasons for excluding articles, identifying how many articles were excluded based on each of the criteria.

Lines 127-130 – Please explain and justify (with references) the choice of parameters for bibliometric analysis in CiteSpace. This explanation will help readers who are not so familiar with bibliometric analysis to understand the process. It will also reinforce the methodological choices.

 Results

Lines 142-144 – Please add a brief explanation of the meaning of modularity value and mean silhouette value. This explanation will help the reader to better understand the data that follows.

Lines 157-158 – It should have a space between the figure’s legend and the following text.

Lines 184-185 – Also, It should have a space between the table’s legend and the following text.

 Discussion

Line 330 - The time limit may imply an internal weakness in the review. As this limitation is neither justified nor clearly addressed, we don't know if there may be studies from before 1995 that are important for the review. I therefore ask the authors to address and discuss this methodological option.

342-371 – This section appears to be a presentation of data, please try to discuss the practical implications for researchers of these "Knowledge structure in terms of journals".

Lines 423-429 – This guidelines are also aligned with ACSM (2021) and the CSEP(2017), please consider the inclusion of this references in order to strengthen the paragraph.

Lines 439-486 – I don't mean to detract from this section, which sums up the issue of inactivity over several disabilities quite well. However, once again, my perception as a reader is that this is a presentation of results. The authors should try to discuss these results by relating them to each other or to other data, for example, their future consequences.

Section 4.2.3. – A very interesting and well-articulated section. In this section, there is no mention of a systematic review of  Mercê et al. (2021) that looks at the barriers and facilitators to practicing the specific physical activity of cycling. This review could have been used in the introduction to frame and reinforce the theme. I suggest reading the review and including it if the authors consider it relevant.

Lines 626-628 – Please explore and discuss this topic a little more.

Lines 647-680 – The authors did a good job of presenting the various scales, but they did not mention their validity/reliability or compare them with each other. This addition of information and comparison would add more depth to the discussion and enrich the manuscript. This in-depth discussion could help readers make decisions about which scales to use in the future, and I therefore recommend this improvement.

Hulteen, R. M., Morgan, P. J., Barnett, L. M., Stodden, D. F., & Lubans, D. R. (2018). Development of Foundational Movement Skills: A Conceptual Model for Physical Activity Across the Lifespan. Sports Medicine, 48(7), 1533-1540. https://doi.org/10.1007/s40279-018-0892-6

Mercê, C., Pereira, J. V., Branco, M., Catela, D., & Cordovil, R. (2021). Training programmes to learn how to ride a bicycle independently for children and youths: a systematic review. Physical Education and Sport Pedagogy, 1-16. https://doi.org/10.1080/17408989.2021.2005014

Reviewer 3 Report

Comments and Suggestions for Authors

This study was a very interesting approach to understanding the literature. It was clearly a LOT of work on the part of the authors. Some of the findings are very interesting and compel the reader to want more information.

Some issues arise as I read this manuscript:

1.        You lump all the physical disabilities together, yet you separate intellectual disabilities, cerebral palsy, and autism. You literally ignore physical disabilities of Deafness or visual impairment. It is hard to believe that in over 1,300 articles you found none with these disabilities. I suggest you revisit the data and separate out these important other physical disabilities.

2.        You count citations of authors, but you do not take into account authors who continually cite themselves. This practice can skew the numbers. Is there a way to determine the true number of citations of other authors?

3.        You literally are reviewing only general journals as you said you only have articles from mainstream general publications. If you could include such journals as Research in Developmental Disabilities, Journal of Visual Impairment and Blindness, British Journal of Visual Impairment, American Annals of the Deaf etc. you would get a much broader set of data with more disabilities represented.

A few other issues:

Pg. 16 line 478 do NOT say healthy peers as having a physical disability does not mean a child is sick. Say peers without a disability.

Pg. 17 Kathleen or Kathllen? This reference is not in the references. Revise

Pg. 18 line 582-583- this is a one-sentence paragraph. Please see APA as you are not following APA 7th edition.

This manuscript has a LOT of random information with very little concrete suggestions. If possible, elaborate on the suggestions for the profession related to your findings.

It may be a good idea to split this into 2 papers. The general overview from the beginning of the paper with not much relevant or useful information (due to the limited number of journals reviewed), and the findings of your studies which is the major important part of the paper with very little on the part of recommendations to professionals.

This paper aims to illuminate physical activity for youth with disabilities, but it seems to raise more questions than it answers.

Round 2

Reviewer 2 Report

Comments and Suggestions for Authors

Dear Authors, 

I congratulate you on your work and thank you for responding to all my suggestions.

After this revision, the manuscript has been strengthened and discussed in greater depth.

Best wishes for continued excellent work.

Reviewer 3 Report

Comments and Suggestions for Authors

This new version is much more meaningful than the first version. Well done.